# Context of Atropine Adherence in Preschool Children with Early-Onset Myopia: A Qualitative Study

**DOI:** 10.3390/children11091087

**Published:** 2024-09-05

**Authors:** Ciao-Lin Ho

**Affiliations:** Department of Child Care and Education, Hungkuang University, No. 1018, Sec. 6, Taiwan Boulevard, Shalu District, Taichung City 433304, Taiwan; swl8310@hk.edu.tw; Tel.: +886-4-26318652 (ext. 5201); Fax: +886-4-26333331

**Keywords:** adherence, myopia, preschool children

## Abstract

The use of atropine is currently one of the most effective methods used to prevent myopia progression. The purpose of this study was to investigate atropine adherence in preschool children with myopia, to explain the context of treatment through caregivers, and to identify barriers and facilitators of using atropine. We conducted in-depth interviews with 60 caregivers of children (parents, kindergarten teachers and nurses) in four different areas ranging from large cities to rural areas in Taiwan. Based on the social ecological theory model, the recorded text was systematically analyzed, extracted, edited and indexed by NVivo 12 Plus. After interviewing caregivers, we determined the barriers and facilitators at the four levels of influence (children, parents, school, and hospital and society). Barriers included the side effects, parental neglect, lack of understanding of long-term drug use, lack of conducive environment, and lack of friendly medical services. Facilitators included overcoming side effects, parental responsibility, myopia progression on treatment adherence, teacher support, management by nurses, navigation by ophthalmologists, and model learning. Hence, establishing a social support network, discussing the experience of individualized drug use in preschool children, and establishing a friendly medical intervention strategy can raise awareness among parents regarding myopia, and improve atropine adherence in preschool children.

## 1. Introduction

Myopia has become an important public health problem, the prevalence of which is rapidly increasing globally, especially in East Asian countries such as China, Hong Kong, Singapore, South Korea, and Taiwan. The prevalence of myopia based on cycloplegic refraction examination among Asian and European schoolchildren (aged 6–19 years) was approximately 60% and 40%, respectively [1]. In Taiwan, the mean ocular refraction of children began to progress into myopia at 11 years old in 1983, and dropped to 8 years old in 2000 [2]. If children show signs of myopia before the age of 6–7, in the absence of proper controls, they may experience high myopia within 7 years and subsequently experience high-myopia-related complications such as retinal detachment, glaucoma and macular degeneration [3,4,5].

Early diagnosis and treatment beginning in preschool have therefore become important issues for children’s health. In addition to correction with glasses, contact lenses, or refractive surgery, the use of atropine is currently among the most effective and cheap methods used to prevent myopia progression [6,7,8]. After 12 months of treatment with 0.01% atropine, the mean myopic progression decreased by 61.9% from 1.05 D/year to 0.40 D/year [9]. Moreover, the combination of 0.01% atropine and orthokeratology was more effective in slowing axial elongation than either alone [10], but orthokeratology is expensive. Low-dose atropine significantly slowed the rate of myopia progression in European pediatric patients with a favorable safety profile [11]. Atropine could significantly slow myopia progression in children, with greater effects in Asian than in white children [12]. Data provided by the National Health Insurance Research Database of Taiwan indicated that the usage rate of the atropine for school-aged children with myopia increased from 36.9% to 49.5% between 2000 and 2007 [13], showing that atropine use in Taiwan has gradually increased.

Atropine is an anti-alkaloid drug extracted from plants and is most commonly used to improve ciliary muscle contraction and cohesion in both eyes [14]. Even though there have been many empirical studies supporting the efficacy of atropine treatment, there are side effects such as photophobia, glare, blurred vision, indistinguishable vision, allergic reactions, allergic conjunctivitis, and elevated intraocular pressure [15]. The rebound effect of the spherical equivalent refractive error and axial length after withdrawal [9] and long-term exposure may cause exposure of the lens and retina to ultraviolet light [16]. In clinical studies, using Atropine once per day at night minimizes the impact on daily school activities for children as the drug’s efficacy decreases during daytime [17]. However, it is worth noting that a high dropout rate of 16–58% was reported; this is due to the photophobia side effect, and constitutes the most important bottleneck in current myopia treatment [18,19].

Past studies have focused only on atropine side effects that result in low treatment adherence [6,9,10], but few studies have explored the possible bottlenecks, opportunities, and core issues that affect caregivers (parents, kindergarten teachers, and kindergarten nurses) in assisting children with atropine treatment.

This study hopes to explore experiences of atropine treatment through in-depth interviews with caregivers of young children with myopia, to improve atropine treatment practice knowledge, and gradually identify barriers and facilitators to atropine adherence in children with myopia based on the four levels of influence—children, parents, school, hospital and society. It is expected that this study will provide a reference for medical practitioners in developing interventional measures, clinical care guidelines, and health policies.

## 2. Methods

### 2.1. Types of Participants

This study was a part of a large-scale group quality study. In order to include different classes and groups, stratified sampling was used to select public and private kindergartens in four areas—northern, central, southern and eastern Taiwan. Through the kindergarten director and teacher, parents, kindergarten teachers and nurses of young children with myopia were invited to participate in this study. Participants could clearly communicate in the Chinese language. Participants who cared for children with orthokeratology or high myopia were excluded. 

### 2.2. Data Collection

The guidelines for semi-structured interviews were prepared for open-ended questions, and interview data were collected from 1 July 2016 to 29 September 2018. We conducted in-depth interviews with caregivers of children (parents, kindergarten teachers, and kindergarten nurses) to understand the adherence to using atropine. Before the official interview began, the researcher sent the interview specifications and outline to each participant by written letter or email. Further, no access issues were raised so face-to-face interviews could proceed smoothly. Each interview took approximately 30–60 min at a mutually agreed upon interview site. There was a brief, one-minute greeting before each interview to build a relationship. The two core questions in the interview guide were: 1. Please share with me your experience of beginning atropine treatment, your thoughts or behaviors at the time of atropine treatment, and any barriers you encountered that led to the subsequent failure to continue treatment. 2. Do you want to help your child continue atropine treatment? What is your experience? What methods will you use? Do you feel supported? To ensure the integrity of interview data, the entire conversation was recorded with two voice recorders, and the verbatim draft was completed within 24 h after the interview.

### 2.3. Interview Data Processing and Analysis

Following the completion of verbatim transcripts, the participants were invited to verify the accuracy of the data via letter or email. Subsequently, the interview data were encoded. A systematic analysis of the verbatim drafts was conducted using a thematic coding approach. Key concepts were identified through keyword searches and analyzed within the context of individual participants and the broader study group. The data were organized into categories aligned with the social ecological framework to identify barriers and facilitators to atropine adherence at the child, parent, school, and hospital and society levels. To ensure the validity of the qualitative data, the Critical Appraisal Checklist was employed to facilitate interrater reliability and consensus on the research findings. The recorded text was systematically analyzed, extracted, edited, and indexed using the qualitative analysis software NVivo 12 Plus.

### 2.4. Research Ethical Considerations

To ensure ethical compliance, the researchers underwent extensive training on institutional review board (IRB) procedures and obtained the necessary certification. The research plan was submitted to the Human Test Committee of Yang-Ming University in Taiwan for review and was approved under reference number YM105073F. Following IRB approval, participant recruitment and interviews were conducted from 30 September 2016, to 29 September 2018.

## 3. Results

A total of 60 participants were recruited from northern, central, eastern, and southern Taiwan (Figure 1). This diverse geographic representation ensured adequate sampling across urban and rural areas. The participant pool included 20 parents, 20 teachers, and 20 nurses, identified by codes A to T (Table 1). In-depth interviews with caregivers (parents, teachers, and nursing staff) were conducted to explore the barriers and facilitators to atropine adherence across four levels of influence: child, parent, school, and hospital and society. The relevant themes and subthemes affecting atropine use in children are summarized in Table 2, based on the participants’ experiences.

### 3.1. Children Level

#### 3.1.1. Barrier Theme: Side Effects of Atropine on the Body and Life

##### Subtheme 1: Side Effects of Atropine

At the outset of treatment, parents often lacked sufficient medical knowledge to understand and address the side effects of the atropine that their children were taking, leading to treatment discontinuation. The use of atropine can have potential consequences for the body and life of young children. Photophobia was the most commonly reported side effect. Atropine paralyzes the ciliary muscle, dilating the pupil and increasing light sensitivity by approximately 25 times. This can cause discomfort or difficulty opening the eyes in bright environments. Blurred vision at distance was another common side effect, requiring children to look at objects from a close distance for clarity. Many parents described a stinging sensation associated with atropine administration, which may have contributed to treatment anxiety and refusal. When children exhibited distress or resistance, parents often became frustrated and decided to discontinue treatment.


*My child complained of severe pain and discomfort, making it difficult to move. The bright light from the sun and shiny cars was overwhelming. Even after the eye discomfort subsided after 3 days, I didn’t dare to use atropine again for my child.*

*(Northern-Parents-M)*



*My daughter, who has −0.5 D vision in both eyes, started using atropine. During her physical education classes, despite wearing hats and glasses, she couldn’t move comfortably. So, the night before, I asked her to discontinue atropine.*

*(Central-Parents-J)*



*My son was uncomfortable and unable to clearly see his homework. Objects nearby appeared blurred, which may have affected his studies.*

*(Eastern-Parents-I)*



*He only said that he didn’t want to use eye drops because he experienced slight stinging in his eyes. The boy next to him also said: I am not willing to use atropine.*

*(Southern-Parents-A)*


##### Subtheme 2: Challenges in Overcoming Side Effects

Lifestyle adjustments were required to manage the side effects of long-term atropine use. The participants reported having to wear hats and sunglasses during outdoor activities.


*The objects in the vicinity appear blurred, affecting learning. Even after discontinuing the medicine, my child continues to experience discomfort. I regret using the eye drops on my child.*

*(Eastern-Parents-K)*



*He should wear sunglasses and a hat, but he prefers not to. At school, he might have experienced eye discomfort, discouraging him from wearing a hat and sunglasses when outdoors. His eyes are sensitive to sunlight, so I am afraid sunlight exposure will exacerbate his symptoms.*

*(Central-Parents-J)*



*The morning after using atropine, my child went out like a donkey, and he couldn’t keep his eyes open while running. My child became extremely sensitive to light. He couldn’t open his eyes even during play activities and photo sessions.*

*(Northern-Parents-H)*


#### 3.1.2. Facilitator Theme: Overcoming Side Effects through Self-Management

Parental guidance and the development of independent daily routines were instrumental in facilitating long-term atropine use. The ability to self-manage side effects was a critical factor in treatment adherence. Parents and children expressed a positive outlook on the treatment’s effectiveness.


*My son is in elementary school. I hope he can manage the treatment independently. I believe they are capable of self-administering atropine.*

*(Northern-Parents-L)*



*My daughter is proactive in using atropine by herself. She takes care of everything herself. When I asked if she used it yesterday, she nodded affirmatively.*

*(Northern-Parents-M)*


### 3.2. Parents Level

Parental decisions considerably influenced the management of myopia in young children. Some parents considered myopia as a minor symptom, requiring only glasses, whereas others lacked awareness of the disease and its potential consequences.

#### 3.2.1. Barrier Theme 1: Parental Neglect

##### Subtheme 1: Myopia Need Not Be Dealt With

Some parents perceived myopia to be a nonurgent condition requiring minimal intervention beyond glasses. This lack of urgency hindered their motivation to pursue myopia correction and often led to inconsistent or ineffective treatment adherence.


*The ophthalmologist said that if we discontinued atropine, we would need to take the medicine again, about once a month. Given our busy schedules, maintaining consistent treatment is challenging, as we often neglect follow-up appointments.*

*(Central-Parents-J)*



*Despite ophthalmologists’ recommendations for atropine use for myopia, many parents still don’t recognize myopia as a disease requiring treatment. Therefore, they are reluctant to bring their children back to the clinic and use atropine.*

*(Central-Nurse-Y)*


Most parents reported that obtaining a single small bottle of atropine per visit led to frequent doctor’s appointments. Even high-income families faced challenges in finding time for regular medical care, resulting in interruptions in treatment.


*Atropine cannot be used for a long time. Due to its limited quantity, frequent refills are necessary, which can be inconvenient.*

*(Northern-Parents-E)*



*Parents here are all engaged in business, so they are sometimes very busy. Due to its small size, the parents are hesitant to take it more than once or twice.*

*(Northern-Nurse-V)*


##### Subtheme 2: Iatrophobia

Some families still had iatrophobia (a fear of doctors), often rooted in cultural beliefs. This fear prevented them from seeking medical treatment for perceived health conditions, including myopia. A father, employed as a public office clerk, favored natural remedies such as outdoor activities over traditional medical interventions for his child.


*I believe it’s best to avoid foreign substances whenever possible. Even when my child has a fever, we prefer to use natural remedies such as exercise instead of seeking medical attention from an ophthalmologist.*

*(Southern-Parents-B)*


#### 3.2.2. Barrier Theme 2: Expecting Immediate Cure and Lack of Understanding of Long-Term Drug Use

The perceived effectiveness of atropine treatment was a key factor in determining treatment adherence. Parents who did not observe immediate improvements were more likely to discontinue the medication.

##### Subtheme 1: Expecting Immediate Cure

The perceived efficacy of atropine treatment influenced parental decisions to continue or discontinue treatment. One mother (J) noted that initial success in controlling myopia progression led to continued adherence. However, when myopia progression persisted, the perception of treatment ineffectiveness led to discontinuation.


*I administered atropine daily to my child and initially observed a positive effect on their myopia. However, when the improvement wasn’t sustained, I questioned the effectiveness of the treatment. I believe that atropine may not be beneficial for myopia management.*

*(Central-Parents-J)*


##### Subtheme 2: Long-Term Atropine Use as a Burden

Most of the respondents expressed concerns regarding the ingredients and potential side effects of atropine. Concerns were raised about the effects of long-term atropine use on children’s lives, including academic performance, daily activities, interpersonal relationships, and overall learning. A mother (O) noted that atropine could lead to prolonged discomfort and difficulty with tasks such as reading and writing, making it unsuitable for long-term use.


*My child experiences discomfort and struggles to clearly see the contents of his homework due to blurred vision. Although I understand that atropine might be used long-term, I have concerns about its continued use.*

*(Northern-Parents-O)*



*After using atropine, my child’s vision became blurry and unclear. Despite researching the medication online, I remain worried about the potential long-term effects on their brain development.*

*(North-Parents-G)*


#### 3.2.3. Facilitator Theme 1: Parental Responsibility for Child Health

Adherence to the ophthalmologist’s recommendations and regular follow-up visits were associated with positive outcomes in myopia treatment. This reinforced parental confidence in the treatment plan and alleviated psychological distress.

##### Subtheme 1: Concern and Responsibility for Children

The diagnosis of myopia often elicited feelings of shock, disbelief, and anxiety among parents. They actively sought out reliable ophthalmologists for treatment. Establishing trust in a competent ophthalmologist provided reassurance and increased willingness to follow treatment recommendations.


*I sought out the best ophthalmologists in the area to address my child’s eye condition. We consulted with three different ophthalmologists, and they all recommended atropine treatment. We were very cooperative and followed their recommendations.*

*(Southern-Parents-A)*


Many parents expressed conflicting feelings regarding treatment. Although untreated myopia could lead to progression, they were also concerned about the potential irreversible side effects of atropine. Therefore, informed decision-making about continued atropine use required a thorough risk assessment. Due to initial uncertainty about myopia treatment, some parents relied on online resources or sought advice from friends to evaluate the potential benefits and risks of atropine.


*I researched extensively on atropine and discovered that it has been banned in some countries for more than a decade. Concerned about the potential risks, I decided to discontinue treatment and avoid further ophthalmology visits.*

*(Southern-Parents-C)*



*I believe atropine is ineffective and harmful. My online research suggests that atropine may not be beneficial for eye health.*

*(Central-Parents-J)*


##### Subtheme 2: Children’s Health as a Top Priority

During the initial stages of atropine treatment, parental concerns centered on the potential consequences of untreated myopia, including retinal detachment. Despite these fears, parents often choose to initiate atropine use as a preventive measure. Recognizing myopia as a disease and understanding its potential complications was a pivotal factor in initiating treatment.


*The ophthalmologist warned that without atropine, my child’s myopia would worsen and may lead to retinal detachment in the future. Despite my concerns and reservations, I decided to start atropine treatment to mitigate these risks.*

*(Southern-Parents-A)*


#### 3.2.4. Facilitator Theme 2: The Effect of Myopia Progression on Treatment Adherence

The perceived effectiveness of atropine treatment was a crucial factor in determining long-time adherence. Parents who observed positive outcomes were more likely to continue the medication. When parents noticed improvements in their child’s myopia, it reinforced their belief in the treatment’s effectiveness, encouraging them to trust the ophthalmologist’s recommendations and continue treatment. The positive outcomes alleviated parental psychological stress and reinforced their commitment to the treatment plan.


*When I discovered that my kindergartener’s older siblings had poor eyesight, I proactively administered atropine to both my children as a preventive measure. After a year of treatment, I’ve observed positive results and believe atropine is effective.*

*(Northern-Parents-H)*



*Following the ophthalmologist’s recommendation, I’ve been using atropine for my son every night. His myopia has been stable for almost a year. Based on this, I feel that atropine treatment is effective.*

*(Northern-Parents-L)*


Consistent daily use of atropine was emphasized, even during holidays or periods of physical or mental discomfort. One father (D) shared his treatment experience and noted that continuous treatment can control myopia. This encouraged him to cooperate with the treatment plan and strive for rapid myopia control.


*My child was initially hesitant to use atropine due to light sensitivity. However, after his myopia progressed from −1.5 D to −2.0 D, he was scared and has been motivated to use atropine for a month. His myopia then reduced to −1.25 D to −1.5 D. Based on this positive experience, I’m hesitant to discontinue atropine treatment.*

*(Southern-Parents-D)*


The regular use of atropine results in more effective control or management of myopia. When parents observed improvements in their child’s myopia, it reinforced their belief in the treatment’s effectiveness, encouraging them to trust the ophthalmologist’s recommendations and continue treatment. The positive outcomes alleviated parental psychological stress and reinforced their commitment to the treatment plan.


*When I discovered that my kindergartener’s older siblings, who attended National Primary School, had poor eyesight, I proactively administered atropine to both my children as a preventive measure. After a year of treatment, I’ve observed positive results and believe atropine is effective.*

*(Northern-Parents-H)*



*My son’s initial diagnosis was pseudomyopia, measuring −0.5 D. We started atropine treatment and noticed improvement. After a month, he reported feeling better, and the doctor confirmed that the pseudomyopia had resolved.*

*(Northern-Parents-F)*



*Following the ophthalmologist’s recommendation, I’ve been using atropine for my son every night. My son’s myopia has now been stable for almost a year, and I feel that atropine treatment is effective.*

*(Northern-Parents-L)*


### 3.3. School Level

To effectively manage myopia in children, parents and caregivers required support from various stakeholders, including nurses, teachers, friends, and relatives. This study explored the social factors influencing atropine use and examined the roles of the medical system, kindergarten nurses, kindergarten teachers, relatives, and friends.

#### 3.3.1. Barrier Theme: Lack of Conducive Environment for Myopia Treatment

In Taiwan’s academic-driven culture, some parents expressed concerns about the potential effect of atropine treatment on their child’s learning and brain development.


*My daughter, who has −0.5 D vision in both eyes, started using atropine. During her physical education classes, despite wearing hats and glasses, she couldn’t move comfortably. So, the night before, I asked her to discontinue atropine.*

*(Central-Parents-J)*



*He should wear sunglasses and a hat, but he prefers not to. At school, he might have experienced eye discomfort, discouraging him from wearing a hat and sunglasses when outdoors. His eyes are sensitive to sunlight, so I am afraid sunlight exposure will exacerbate his symptoms.*

*(Central-Parents-J)*


#### 3.3.2. Facilitator Theme 1: Teacher Support for Atropine Adherence

In Taiwan’s educational context, children spend considerable time in kindergarten. Kindergarten teachers serve as influential role models, and their guidance is valuable to children. Children often follow their teachers’ instruction, viewing teachers as team leaders. The presence of social support networks, such as teachers, is a key factor in myopia prevention. Collaborative efforts with teachers can further enhance myopia prevention systems.


*We’re always willing to assist children if their parents have specific requests. The teachers agreed to the use of atropine, and the children have been compliant. To minimize discomfort, we ensure the classroom is dimly lit, and we avoid outdoor activities during treatment. The side effects are mild, likely just a slight sting or two.*

*(Northern-Teacher-R)*



*The child’s mother expressed concerns about other people administering eye drops, but we’ve been able to assist with the treatment. The discomfort is minimal, only a slight sting or two.*

*(Northern-Teacher-P)*


#### 3.3.3. Facilitator Theme 2: Case Management by Nurses

Kindergarten nurses play a pivotal role in myopia prevention. The interviewed nurses indicated their involvement in conducting multiple health education sessions for children and parents regarding the long-term use of atropine. These sessions focused on providing advice, guidance, and support to enhance knowledge and attitudes. Additionally, case tracking and monthly vision reviews were implemented to facilitate ongoing treatment and monitoring.


*For visually impaired students, the school emphasizes the importance of avoiding excessive light exposure and consistent atropine use through informational leaflets with guidelines for proper use. Regular health education announcements are provided to both students and parents each semester. Case tracking is also conducted, and monthly vision checks are conducted at the health center.*

*(Northern-Nurse-W)*


### 3.4. Hospital and Society Level

During interviews, parents emphasized the need for comprehensive support, encompassing medical treatment and drug delivery. Media outlets, friends, and relatives were identified as influential sources of support. Additionally, the role of various caregivers, including school teachers and nurses, in providing education and collaboration was highlighted. By analyzing interview data, we identified the factors influencing atropine use. The medical system, physicians, kindergarten nurses, kindergarten teachers, media, relatives, and friends were both obstacles and facilitators.

#### 3.4.1. Barrier Theme: Lack of Friendly Medical Services

The lack of personalized instruction and collaboration from physicians is another area of concern. Many parents expected physicians to clearly inform them of the importance, benefits, and care considerations of atropine treatment.


*The medical system often fails to educate parents on preventing myopia progression and cooperating with treatment. Many parents feel that after receiving a diagnosis and prescription, the doctor’s involvement is limited. There’s a lack of communication regarding the importance of eyesight care and the potential side effects of medications. The ophthalmic medical department should improve follow-up care for myopia patients. Doctors should provide parents with guidance on effective treatment strategies to ensure their child’s long-term eye health.*

*(Northern-Nurse-Ding)*


#### 3.4.2. Facilitator Theme 1: The Role of Ophthalmologists as Navigators

Ophthalmologists played a vital role in providing guidance and education. When ophthalmologist recommended atropine use to young children, they were more likely to comply. By emphasizing the positive benefits of atropine, ophthalmologists could alleviate parental concerns and encourage treatment adherence.


*Children heed the doctor’s recommendations over parental advice. During eye exams, the doctor advised my child to use atropine to prevent myopia progression. Upon returning home, I assisted with the medication, and my child felt obligated to comply.*

*(Central-Parents-J)*


#### 3.4.3. Facilitator Theme 2: Model Learning from Significant Others

In addition to medical advice, parents often sought guidance from relatives and friends regarding myopia treatment. By sharing personal experiences, significant others could provide a more relatable perspective, reinforcing parents’ perception of the effectiveness of atropine and encouraging continued use.


*My son has been using atropine for more than 2 years, and his myopia has stabilized at −3.0 D. Encouraged by relatives who own an optical shop, I’ve continued atropine treatment. Even after my child transitioned from elementary to middle school, I’ve maintained the medication because of its positive effect on myopia control.*

*(Northern-Parents-I)*


## 4. Discussion

The onset age of myopia among Taiwanese children is exhibiting a yearly decline, accompanied by a rise in high myopia incidence. Atropine is currently the most effective method for treating myopia; however, atropine is limited by challenges in adherence due to its perceived limited benefits and the complexities of long-term use for both children and parents. This study explored the barriers and facilitators influencing adherence from the perspective of caregivers across four levels: child, parent, school, and hospital and society.

Atropine’s side effects include pupil enlargement, photophobia, discomfort, blurred vision, stinging, constipation, systemic allergies, and itchiness. Medicinal side effects are key contributors to nonadherence [9,10]. Some children may develop a fear of light but may not like to wear sunglasses, leading to refusal to use atropine [19,20,21], thereby imposing a psychological burden on parents [22]. A study revealed that the adherence to occlusion therapy in combination with atropine use decreased over time, reaching 30% after 100 days of treatment [6]. Evidently, physical and mental problems affect the sustainability of drug treatment. The child-level barrier identified in the study is the effect of atropine’s side effects on physical and emotional well-being, which can influence parental decisions. Common side effects include dry eye, allergic conjunctivitis, flushing, headache, and cardiac and urinary symptoms [17]. Photophobia, the most common side effect, is positively correlated with atropine concentration [19]. Children experiencing near-blurred vision may benefit from bifocal or multifocal glasses. To mitigate photophobia symptoms during outdoor activities, hats, photochromic lenses, or sunglasses are recommended. Parents should guide children in developing self-management skills to overcome side effects and adapt to necessary lifestyle changes. Myopia treatment is similar to chronic disease treatment, requiring children to cultivate self-efficacy and long-term adaptation habits. By learning to cope with side effects, children can successfully continue atropine treatment. The study demonstrated that children can transition from initial discomfort to a more adaptive approach, ultimately leading to sustained atropine use.

As primary caregivers, parents play a pivotal role in children’s myopia treatment. Their perception of myopia as a disease affected their willingness to seek medical attention. A study revealed that only 46% of parents recognized myopia as a health risk for their children [20]. To enhance parental knowledge, various strategies can be implemented. For example, the World Health Organization classified myopia as a sensory organ disease in 2004 [21], and the Health Promotion Administration of Taiwan issued a similar statement in 2017. Iatrophobia (the fear of doctors) can discourage parents from seeking medical treatment and can profoundly affect their choice of myopia treatment. Obviously, parental attitudes toward myopia treatment can evolve over time, influenced by factors such as treatment outcomes and personal beliefs. Adherence to atropine treatment varied among parents and children based on their knowledge, awareness, attitude, and actions in response to perceived risks and uncertainties [22]. Personal choice depends on knowledge and affects whether myopia is treated. Parents who are concerned about myopia and its severity are more likely to actively seek information, consult medical professionals, and pursue quality care [23]. Recognizing parental responsibility for their child’s health, some parents use online resources and social media to gather information on treatment options [24]. However, some may be susceptible to misinformation or exaggerated claims circulating online, potentially influencing their health decisions.

Children and adolescents with myopia often experience higher levels of psychological and social stress compared with their emmetropic peers, leading to feelings of inferiority, low self-esteem, low confidence, depression, and sleep difficulties [25,26]. Children with myopia may also be easily distracted, have difficulty learning, and exhibit below-average academic performance, requiring additional support in school and at home [26,27]. Atropine’s side effects, such as anxiety and sleep disturbances, can further affect school performance and social integration. Students who feel unsupported by their teachers may develop negative perceptions of school [28]. The school environment plays a crucial role in myopia prevention. Interventions that promote outdoor activities, physical activity, and reduced near work are recommended [29]. These factors are mutually exclusive with atropine’s side effects, emphasizing the need for comprehensive myopia prevention policies that integrate both approaches. Despite challenges, schools can play a positive role in facilitating atropine adherence. Teachers can educate students about the importance of treatment and provide guidance on proper atropine use. Additionally, school nurses can conduct health education sessions for both children and parents, offering advice, guidance, and support. By fostering a supportive school environment and providing comprehensive education, schools can contribute to successful myopia prevention and treatment.

Navigating the healthcare system can present challenges for parents managing their child’s myopia, including taking time off work, scheduling appointments, communicating with doctors, finding transport, locating the hospital, and traveling long distances [23,30,31]. These factors can hinder parental decision-making [31]. Medical staff in clinics may not have time to provide parents with comprehensive atropine treatment instructions. Parents may not fully understand the importance of continued treatment and the potential consequences of discontinuation, leading to treatment interruptions. Our findings highlight parental concerns about the side effects of atropine and emphasize the need for ophthalmologists to offer clear treatment explanations and lifestyle recommendations [24]. However, busy ophthalmologists may not have time for detailed consultations. Fortunately, social networks can provide valuable support, including guidance from friends, relatives, school nurses, and community members. School nurses can contribute to health education, follow-up management, and regular vision examinations [32]. However, social network support may not be sufficient to address all parental needs. Some parents may feel overwhelmed by the challenges of the treatment process. Further research is necessary to explore strategies for enhancing social network support and improving myopia prevention outcomes.

## 5. Conclusions

This study, grounded in the social ecological theory model, identified barriers and facilitations influencing atropine adherence among caregivers of children with myopia. Through interviews, the study compiled a comprehensive list of barriers and facilitators of the four levels: child, parent, school, and hospital and society. Barriers included side effects on the body and life, parental neglect, misconceptions about long-term drug use, conducive environmental factors for myopia treatment, and a lack of friendly medical services. The facilitators identified included manageability of side effects, parental responsibility, the perceived threat of myopia progression, teacher support, nurse management, physician guidance, and model learning. The study findings suggest enhancing parental education regarding disease awareness and the severity of myopia. Leveraging internet-based interventions can establish a reliable social support, given the ease of accessing information online. The medical system should adopt a patient-centered approach to facilitate atropine use and provide clear explanations of empirical differences among various myopia treatments and potential side effects, fostering greater trust and communication. Furthermore, ophthalmologists, kindergarten nurses, and teachers play crucial collaborative roles. Implementing on-the-job training programs can promote mutual communication and collaboration among these stakeholders.

## Figures and Tables

**Figure 1 children-11-01087-f001:**
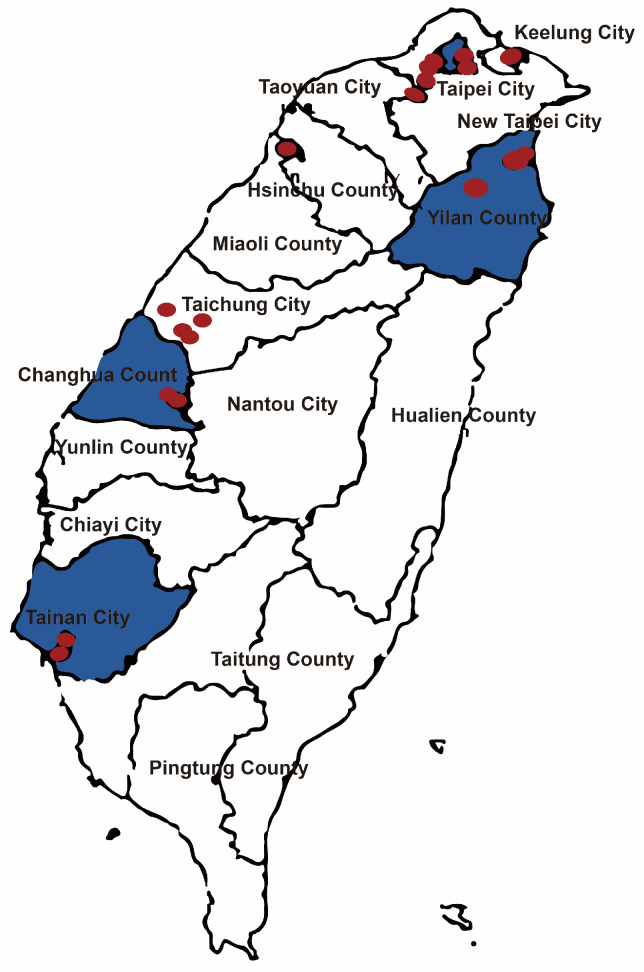
List of interviewed kindergartens.

**Table 1 children-11-01087-t001:** General characteristics (n = 60).

Characteristics	Categories	n (%)
Gender	Female Male	55 (92) 5 (8)
Age (year)	20–30 30–35 36–40 41–45 46–50 Over 51 years old	3 (5) 11 (18) 16 (27) 14 (23) 10 (17) 6 (1)
Education	University High school	59 (98) 1 (2)
Type of work	Kindergarten teacher Nursing staff Home management Business Education Government employee Engineering Baker Freelance	20 (33) 24 (40) 2 (3) 2 (3) 8 (13) 1 (2) 1 (2) 1 (2) 1 (2)

**Table 2 children-11-01087-t002:** Barriers and facilitators for children using atropine.

Levels of Influence	Barriers/Facilitators Themes	Explanation/Example Sub Themes
Children Level		
Barriers	Side Effects of Atropine on the Body and Life.	Side Effects of Atropine Challenges in Overcoming Side Effects
Facilitators	Overcoming Side Effects Through Self-Management	
Parents Level		
Barriers	Parental Neglect	Myopia Need Not Be Dealt with Iatrophobia
	Expecting Immediate Cure and Lack of Understanding of Long-Term Drug Use	Expecting Immediate Cure Long-Term Atropine Use as a Burden
Facilitators	Parental Responsibility for Child Health	Concern and Responsibility for Children Children’s Health as a Top Priority.
	The Effect of Myopia Progression on Treatment Adherence	
School Level		
Barriers	Lack of Conducive Environment for Myopia Treatment	
Facilitators	Teacher Support for Atropine Adherence Case Management by Nurses	
Hospital and Society Level		
Barriers	Lack of Friendly Medical Services.	
Facilitators	The Role of Ophthalmologists as Navigators	
	Model Learning from Significant Others	

## Data Availability

Data are contained within the article.

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
