# Peer review of "Context of Atropine Adherence in Preschool Children with Early-Onset Myopia: A Qualitative Study"

_children, 2024, doi:10.3390/children11091087_

Round 1

Reviewer 1 Report

Comments and Suggestions for Authors

1
The manuscript touches on an important and interesting topic: compliance of children with use of atropine dropps. However, there are several points in the manuscript , which should be reconsidered
In my opinion, the analysis and presentation of the data would require a more concise and scientific approach. I recommend that the author would consult ophthalmology and or epidemiology to better highlight the issue
A few examples of repair needs
The abstract deals more about the methods, results and conclusion would be needed. The abstract, as whole should be more specific.
The study has been conducted in 4 different areas In Taiwan. It would be worth clarifying in more detail in which respects these regions differ from each other or are homogeneous when dealing this issue
Introduction L 27
In Taiwan, ocular refraction in the mean began to progress to a myopic condition in 11 year
old children in 1983 and in 8 year old children in 2000 1
This is either a misinterpretation or an inappropriate reference
L 29 33 the global prevalence is prediction, not a fact
L37, retard myopia .. to prevent or decrease progression
L38. decrease 61 9% compared with prior to treatment) 9 not a good reference or interpretation.... Progression is generally not linear, thes prectages do not say anything.
L 40 ref 10 is not necessary in this context.
L45 the usage rate of myopia treatment for school age children increased from 36 9% to 49 5
use of what
L51 There are even media reports ?
L76Those who refused to participate in the study were excluded
Perhaps flow chart could clarify
Texts for tables and figures ought to be. clearer.
Tables are inaccurate
As a summary, the manuscript should summarize more precisely to explain which were the main reasons for good and bad compliance,

Author Response

Comments 1. The manuscript touches on an important and interesting topic: compliance of children with use of atropine dropps. However, there are several points in the manuscript , which should be reconsidered. In my opinion, the analysis and presentation of the data would require a more concise and scientific approach. I recommend that the author would consult ophthalmology and or epidemiology to better highlight the issue

Response 1: Thanks for the suggestion, we had consulted the ophthalmologist. The reviewer’s comments and modifications will be answered below.

Comments 2. A few examples of repair needs: The abstract deals more about the methods, results and conclusion would be needed. The abstract, as whole should be more specific. The study has been conducted in 4 different areas In Taiwan. It would be worth clarifying in more detail in which respects these regions differ from each other or are homogeneous when dealing this issue

Response 2: The abstract is limited to no more than 200 words in total, so it has been kept as brief as possible. We had revised the description: “The use of atropine is currently the most effective methods to prevent myopia progression. The purpose of this study was to investigate the adherence of using atropine in preschool children with myopia, to explain the context of treatment through caregivers, and to identify barriers and facilitators of using atropine. In-depth interviews with 60 children caregivers (parents, kindergarten teachers and nurses) in four different areas ranging from large cities to rural areas in Taiwan. Based on the social ecological theory model, the record text was systematically analyzed, extracted, edited and indexed by NVivo 12 Plus. After interviewing with caregivers, the study compiled the barriers and facilitators of the four levels (Children; Parents; School; Hospital and Society). Barriers included the side effects, parents neglect, lack of the concept of long-term drug use, lack of an effective environment, and lack of individual friendly medical services. Facilitators included side effects can be overcome, parents' responsibility, myopia progression make parents agreement, teacher’s help, nurses’ management, doctor’s navigation, and model learning. Hence, establishing a social support network, discussing the experience of individualized drug use in pre-school children, and establishing a friendly medical intervention strategy can aware parents' myopia insight and improve atropine adherence for preschool children.”

Comments 3. Introduction: L 27, In Taiwan, ocular refraction in the mean began to progress to a myopic condition in 11 year old children in 1983 and in 8 year old children in 2000 [1]. This is either a misinterpretation or an inappropriate reference

Response 3: We had revised the description: Line 30-31 “In Taiwan, the mean ocular refraction of children began to progress into myopia when they were 11 years old in 1983, and dropped to 8 years old in 2000 [2].”

Comments 4. L 29 33 the global prevalence is prediction, not a fact

Response 4: We had revised the description: Line 28-30 “The prevalence of myopia based on cycloplegic refraction examinations among Asian and European schoolchildren (aged 6–19 years) was approximately 60% and 40%, respectively [1].”

Comments 5. L37, retard myopia .. to prevent or decrease progression

Response 5: We had revised the description: Line 39 “In addition to correction with glasses, contact lenses, or refractive surgery, the use of atropine is currently among the most effective and cheap methods to prevent myopia progression [6–8].”

Comments 6. L38. decrease 61 9% compared with prior to treatment) 9 not a good reference or interpretation.... Progression is generally not linear, thes prectages do not say anything.

Response 6: Progression is generally not linear, but it can be shown that the myopia progression is slower with the use of atropine. We had revised the description: Line 39-41 “After 12 months of treatment with 0.01% atropine, the mean myopic progression decreased by 61.9% from 1.05 D/year to 0.40 D/year [9].”

Comments 7. L 40 ref 10 is not necessary in this context.

Response 7: This statement indicates that atropine and orthokeratology are two effective treatments for myopia, so it is recommended to retain them. We had revised the description: Line 41-43 “Moreover, the combination of atropine 0.01% and orthokeratology was more effective in slowing axial elongation than anyone of combination [10], but orthokeratology is expensive.”

Comments 8. L45 the usage rate of myopia treatment for school age children increased from 36 9% to 49 5 use of what

Response 8: We had revised the description: Line 46-49 “Data provided by the National Health Insurance Research Database of Taiwan indicated that the usage rate of the atropine for school-age children with myopia increased from 36.9% to 49.5% between 2000 and 2007 [13], showing that atropine use in Taiwan had gradually increased.”

Comments 9. L51 There are even media reports ?

Response 9: We had deleted “There are even media reports that atropine increases the risk of glaucoma, causing concern and rejection among children and parents, which often leads to drug non-compliance.”

Comments 10. L76Those who refused to participate in the study were excluded
Perhaps flow chart could clarify
Response 10: We had revised the description: Line 79-80 “Participants who cared for children with orthokeratology or high myopia were excluded.”

Comments 11. Texts for tables and figures ought to be. clearer.
Response 11: The text of tables and figures had been revised for clarity. If the formatting error may be caused by the word version, it is recommended to look at the PDF file.

Comments 12. Tables are inaccurate
Response 12: The format of table 1 and table 2 had been modified. If the formatting error may be caused by the word version, it is recommended to look at the PDF file.

Comments 13. As a summary, the manuscript should summarize more precisely to explain which were the main reasons for good and bad compliance,

Response 13: We had revised the description: Line 516-532 “This study indicated barriers and facilitations of using atropine based on the social ecological theory model. After interviewing with caregivers, the study compiled the barriers and facilitators of the four levels (Children; Parents; School; Hospital and Society). Barriers included the side effects on children body and life, parents neglect, lack of the concept of long-term drug use, lack of an effective environment for myopia treatment, and lack of individual friendly medical services. Facilitators included side effects can be overcome, parents' responsibility, myopia progression make parents agreement, the teacher’s help, the nurses’ management, doctor’s navigation, and model learning from significant others. This study suggests a strengthening of parents’ education in regard to disease awareness and the severity of myopia. Establishing a reliable social support by Internet intervention strategy may be an opportunity because the Internet is the easiest way to get information. The medical system should develop a friendly approach to assist in atropine use and strengthen the description of empirical differences in var-ious myopia treatments as well as of side effects to provide parents with more choices and increase communication and trust. In addition, physicians, kindergarten nurses and teachers play an important collaborative role, suggesting the development of on-the-job education to promote mutual communication and collaboration.”

Reviewer 2 Report

Comments and Suggestions for Authors

The discussion about Atropin usage for myopia control is very popular in scientific community and between ophthalmology experts. Usually scientific interest is mostly focused on effect and side effects of the Atropin. However, since the duration of the therapy is usually longer, compliance of the child and caregiver has a important role. In this paper focus is on the social effects of the treatment which is interesting and can be helpful. The paper is well structured, sufficiently informative, with well defined structure. Results are clearly presented.

Author Response

Comments 1. The discussion about Atropin usage for myopia control is very popular in scientific community and between ophthalmology experts. Usually scientific interest is mostly focused on effect and side effects of the Atropin. However, since the duration of the therapy is usually longer, compliance of the child and caregiver has a important role. In this paper focus is on the social effects of the treatment which is interesting and can be helpful. The paper is well structured, sufficiently informative, with well defined structure. Results are clearly presented.

Response 1: Thank you for the comments concerning our manuscript.

Reviewer 3 Report

Comments and Suggestions for Authors

Extremely relevant work, easy reading and comprehension.

With the increase in the use of screens, cell phones, tablets and laptops, especially after the Covid-19 pandemic, the number of children with myopia has grown every year.

In the introduction, I suggest including data on the number of children with myopia, and data from previous studies on the effectiveness of atropine use in an attempt to stop the development of the pathology.

Also in the introduction, I missed possible interventions regarding the time of application of the eye drops, how many times a day it is necessary to use them. If it is done once a day, if the child uses it before bedtime, will it have the benefit, since during the day, it can affect school activities? 

If these data exist in the literature, I believe they should be included in the manuscript. 

Is there another pharmacological option or only atropine?

I believe the table 1 is not formatting, adjust because it is hard to understand

Author Response

Comments 1. In the introduction, I suggest including data on the number of children with myopia, and data from previous studies on the effectiveness of atropine use in an attempt to stop the development of the pathology.

Response 1: We had revised the description: Line 28-30 “The prevalence of myopia based on cycloplegic refraction examinations among Asian and European schoolchildren (aged 6–19 years) was approximately 60% and 40%, respectively [1].” and Line 39-41 “After 12 months of treatment with 0.01% atropine, the mean myopic progression decreased by 61.9% to 0.40 D/year compared with before treatment [9].”  

Comments 2. Also in the introduction, I missed possible interventions regarding the time of application of the eye drops, how many times a day it is necessary to use them. If it is done once a day, if the child uses it before bedtime, will it have the benefit, since during the day, it can affect school activities?  If these data exist in the literature, I believe they should be included in the manuscript. 

Response 2: We had revised the description: Line 56-58 “In clinical studies, using Atropine once per day at night minimizes the impact on daily school activities for children as the drug's efficacy decreases during daytime [17].”

Comments 3. Is there another pharmacological option or only atropine?

Response 3: Myopia is generally treated with atropine or orthokeratology. Pharmacological option is only atropine.

Comments 4. I believe the table 1 is not formatting, adjust because it is hard to understand

Response 4: The format of table 1 and table 2 had been modified. If the formatting error may be caused by the word version, it is recommended to look at the PDF file.

Round 2

Reviewer 1 Report

Comments and Suggestions for Authors

This is imprtant and topical article of the practical questions relted to use of atropine drops in children.